# Specters of Mob in David Cronenberg's *Eastern Promises*

## Gerd Bayer

Department of English and America Studies, Friedrich-Alexander-Universität Erlangen-Nürnberg, 91054 Erlangen, Germany; gerd.bayer@fau.de

**Abstract:** This article situates David Cronenberg's film *Eastern Promises* in the context of post-Cold-War European narratives. It argues that the secret dealings of the Russian mob in London are presented in the film as the uncanny and spectral return of forms of government and business that run counter to the rationale conventionally associated with democratic capitalism and at the same time reveal much about its inherent logic. Cronenberg's film connects private traumata with the violent reality of globalization, staging one as the ghostly realization of the other.

**Keywords:** Cronenberg; David; *Eastern Promises*; Derrida; Jacques; spectrality; Russian mob

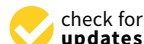



## 1. Introduction

David Cronenberg's cinema has consistently addressed matters of technology and the human, the interplay of bodily desires and the fascination that machines afford in the pursuit of such happiness. When corporeality hits against materiality, this frequently results in painful violence, and while earlier Cronenberg films such as *Videodrome* (1983) and *Crash* (1996) may have situated this encounter squarely in a landscape of technological saturation, films like *A History of Violence* (2005) have followed up on this mode within rural settings and alongside plots of romantic entanglements. In his BBC film *Eastern Promises* (2007), Cronenberg once again reconnects with his central motifs yet moves into an environment that, though urban, is less determined by the fetishization of technology that had turned into a trademark of Cronenberg movies. This film is instead set in the criminal and highly violent world of the Russian mob, called vory v zakone, as it draws money out of restaurants and forced prostitution in London. When a pregnant teenager delivers her baby shortly before dying, Anna, the midwife at the clinic (played by Naomi Watts), takes the mother's Russian diary home to relocate the baby with its family but is only able to connect, without realizing which dangerous path she is following, the deceased with her very tormentors, Russian restaurant owner Semyon (Armin Müller-Stahl), his mentally somewhat unstable son Kirill (Vincent Kassel), and their driver Nikolai (Viggo Mortensen), who is revealed at the end of the film to be a Russian secret agent reporting to British Intelligence. It is this double existence of one of the film's protagonists that provides a gateway to read *Eastern Promises* as a ghostly encounter with matters of organized crime, of the status of migrant laborers, and with the role assigned to ethnic identities in Europe's large metropolises during the early decades of the twenty-first century. As Scott Wilson suggests, "*Eastern Promises* establishes a set of conflicting positions regarding emigration and disappointment, the law and its obscene underside, the vanishing past of a lost homeland and the new, violent traditions of a contemporary existence" (Wilson 2011, p. 84) By toying with the uncanny nature of (post)national existence and morality, Cronenberg's film, this essay argues, draws on the spectral quality that attaches to some of the major narratives of contemporary identity politics.

A focus on the spectral features of this film nevertheless provides various points of contact to established scholarly discussions relating to Cronenberg's films. Film historians will quickly note the prominent noir overtones in *Eastern Promises* and the film's commitment to stark lighting, a criminal milieu, and the moral complexity of its main

characters, resulting in what Sue Short has described as "a tale of two cities" that relies on "a series of fronts" behind which characters retreat in order to perform a more secretive professional, criminal, or sexual identity (Short 2019, p. 108). Even when seen as a neo-noir film, *Eastern Promises* nevertheless surprises by its rather idiosyncratic use of the detective, who remains undercover pretty much throughout the film, and by its overall refusal to endow its female characters with the necessary quantity of agency and general aplomb, despite the force that attaches to Naomi Watts's character. A further link to noir traditions exists through Cronenberg's omnipresent use of technology throughout his oeuvre. In the same way that modernist filmmakers such as Walter Ruttmann and Dziga Vertov had visually toyed with a "fusion of man and machine" (Strathausen 2003, p. 30), Cronenberg presents Viggo Mortensen as intimately linked to his job as driver and Naomi Watts as tied in a nostalgic fascination to her Russian motorcycle, an Ural Solo. Finally, the noir connection also exists through the visual language created by *Eastern Promises*, resulting from Cronenberg's cooperation with cinematographer Peter Suschitzky, who has such diverse films to his credit as *The Rocky Horror Picture Show* (1975) and the Star Wars episode *The Empire Strikes Back* (1980), and who has been Cronenberg's cinematographer since *Dead Ringers* (1988). Duncan Petrie notes about Suschitzky's style that it relies on "an expressionist impulse" built on "light and shade and colour" (Petrie 1996, p. 142), qualities that underline a stylistic proximity for a neo-noir visual aesthetics. Lastly, one could also point to the script for this film, contributed by Steven Knight, whose other credits also include Stephen Frears's *Dirty Pretty Things* (2002), another London-based film that has clear neo-noir overtones. There are, then, clear connections to the historical noir tradition also in *Eastern Promises*, yet they benefit from the reading presented in this essay of the uncanny nature of the film's representation of social realities in that they constitute both continuation within and departure from the central concerns of noir filmmaking.

The uncanny in *Eastern Promises* also builds on Cronenberg's general fascination with bodily mutilations, excessively violent scenes, the introjection of technology into living organisms, or the way in which seemingly harmless and peaceful environments suddenly disintegrate and reveal some underlying horrific aspect of a particular society. The film follows a trajectory in Cronenberg's changing investment in corporealities, also visible in his *A Dangerous Method* (2011), with its prominent A-list Hollywood cast, that invites a rethinking of Cronenberg as the schlocky Canadian art-house filmmaker, which he clearly represented in his earlier films, in the light of his success on the global and commercial film scene. Criticism has already begun to discuss the notion that Cronenberg's "films, however disruptive of genre conventions, are distributed within the mainstream media marketplace and must be understood as at least partially determined by these conditions even as they expand and call into question these limits" (Varga 2003, p. 268). *Eastern Promises* clearly falls within these categories, yet its uncanny and ghostly qualities nevertheless also hark back to vintage Cronenberg, translating gory special effects into high-resolution realism, for instance in the scene where Nikolai disembodies a corpse by removing individual fingers with medical instruments. While earlier Cronenberg films have turned machines into threatening foes, *Eastern Promises* relies on freezers and hospital machinery to be functional and efficient, moving the violence onto an inter-human plane.

For these and other reasons, Cronenberg remains a filmmaker whose output is not easily categorized, possibly resulting from his own particular personal and aesthetic socialization. He comes from a Lithuanian Jewish background, with both his parents already born in North America, and one could say of his commitment to Jewish identities what Frederic Raphael has reported of Stanley Kubrick, namely that he supposedly said about himself that "he was not really a Jew, he just happened to have two Jewish parents" (Raphael 1999, p. 108). Both filmmakers, all the same, share an interest in uncanny cinematic spaces, or rather what Mark Windsor has termed "classic uncanny stories" (Windsor 2019, p. 52). Additionally, possibly harking back to his diasporic family legacy, Cronenberg's films often feature "outsider characters in central roles" (Morris 1994, p. 20), the kinds of uncanny identities that also mark the main roles in *Eastern Promises*. Carsten

Strathausen, in discussing early cinema's representation of urban spaces by Ruttmann and Vertov, draws a comparison between "[m]odernity's oscillation between exposure and repression, between location and displacement" (Strathausen 2003, p. 15) and the Freudian notion of the uncanny. What he diagnoses for early twentieth-century metropolises such as Berlin and for their citizens' experience of "spatial incoherence and disorientation" (Strathausen 2003, p. 22) also applies to the post-Wall urban migrant. Indeed, as Peter Morris's biography of Cronenberg the filmmaker suggests, it is precisely this duality of gesture that marks his cinematic oeuvre, the fact that it sets out to bring together supposedly contradictory impulses such as "detachment and passionate engagement" (Morris 1994, p. 130). Inscribed into these biographical aspects of Cronenberg's work and into the conflicts his filmic characters encounter are precisely the spectral qualities that *Eastern Promises* so prominently displays. Society itself, so massively re-shifted in the aftermath of World War II and affected by the forces of globalization, has become ghostly, or so one could argue.

Nadia Butt comes to a similar conclusion when she observes that the Canadian filmmaker routinely creates narratives about failed integration, where "Cronenberg addresses specifically the predicament of those who are 'stuck'" (Butt 2010, p. 256). As Mike Mason argues with respect to the "essential narrative problematic in social-realist works", the protagonist frequently finds him- or herself in a "conflict of loyalties in relation to particular, clearly defined communities" (Mason 2001, p. 248). Writing about Nikolai's character, Aron Dunlap and Joshua Delpech-Ramey detect "a mixture of humorous detachment and slapstick chagrin" (Dunlap and Delpech-Ramey 2010, p. 322), while Cynthia Freeland acutely observes that he is a person "who seems to have erased or denied his own identity" (Freeland 2012, p. 34), making of him the prototypical representative of the film's overall take on shifting identities. It is precisely this sense of being torn between identities and affiliations that generally marks *Eastern Promises* and the way it presents Russian–British identities in turn-of-the-millennium London, presented in Cronenberg's film as a criminal and spectral city.

## 2. Ghosts of the Past, Ghosts of the Present

As the title of this essay already suggests, this reading of the film draws from a specific textual tradition, one whose trace leads from Cronenberg to Derrida's reading of Marx, seeing in all of these textual and cinematic discourses the ghostly presences or absences of Freud's writing on the uncanny, who—to follow Nicholas Royle—"keep trying to lay certain ghosts to rest, but they keep coming back" (Royle 2003, p. 51). Mark Windsor has recently observed about uncanny cinematic works that they frequently "create the dubious appearance of the supernatural in the context of one's experience of reality" (Windsor 2019, p. 65) and in this phrase succinctly summarizes one of the central concerns that almost all of Cronenberg's films address. In other words, this essay follows the ghost of an idea that sees in the bodily instability of the film's characters, in the subliminal threat of violence's omnipresence, and in the suppressed memories of earlier traumas a return to cultural ghosts. These are the very ghosts that Francis Fukuyama, following the fall of the Berlin Wall, had famously pronounced gone, along with the end of history. At that precise historical moment, as Jacques Derrida argues in *Spectres of Marx*, the necessity to reflect on the afterlife of Marxism took on spectral dimensions. Recently thrown onto history's junkyard, Marxism returns from the dead as a ghost, thus taking on, rather appropriately, the very form of its first appearance. It thereby evokes the spectacular beginning of *The Communist Manifesto* and its notoriously literary, if not to say gothic, opening, with the lines "Ein Gespenst geht um in Europa" (Marx and Engels 1848, p. 1). Like any decent ghost, this one too is prone to return to its original haunts, and, in the setting of Cronenberg's film, the ghost, first un-seen by Marx and Engels, returns to London.

The London Cronenberg shows in his film takes on spectral qualities itself. As the film critic for the *New York Times* noted, *Eastern Promises* gives London a "sinister, palpitating presence" (Scott 2007), alluding to the Gothic and ghostly spatial presence of the British capital. That the film's aesthetic invites such a metaphorical decoding is echoed by another

newspaper review, this one published in the *Houston Chronicle*, uncannily on the same day, 14 September 2007, that also saw the *New York Times* review. In this write-up, the critic notes that the film is not really "about Russian gangs so much as Cronenberg's own dark passions". Both reviewers took note of the fact that the film should not be read through its surface meaning but instead invites a subconscious response, one that pays attention to its ghostly aesthetic and its subliminal psychic dynamic. Without having said so themselves, both reviewers described *Eastern Promises* as an uncanny film, as a cinematic example of what Freud describes as Das Unheimliche, the unsettling and disturbing experience of a moment that is simultaneously both familiar and somewhat strange. They thus agree with an assessment by Dunlap and Delpech-Ramey, who see in *Eastern Promises* (and *A History of Violence*) a turn towards a version of the grotesque that they describe as "an uncanny *internal* force" (Dunlap and Delpech-Ramey 2010, p. 328) and trace to Nikolai's subverted notion of male heroism. In terms of how Cronenberg appropriates London in his film, he almost entirely avoids the well-known tourist images, presenting instead an image of London that rests on the city's lesser-known social and ethnic spheres. At the time of Cronenberg's filming of *Eastern Promises*, London was still in the process of coming to terms with what Tony Fitzmaurice has described as "the realities of social exclusion and alienation in the Thatcher era" (Fitzmaurice 2001, p. 27). It is particularly through the setting of the film, its grimy localities, that these non-glitzy aspects of London take on center stage, reducing iconic landmarks (such as, for instance, Tower Bridge) to brief glimpses caught in the background of a tracking shot that focuses on a more mundane foreground.

To tease out some of these specters, let me turn to a few stills from the film to show how Cronenberg stages London. One type of setting in the film relates directly to the private lives of the protagonists such as, for instance, Anna's private home and place of work or Semyon's restaurant. These settings evoke homeliness and cozyness, held in dark colors and lavishly decorated. One scene at the Russian restaurant lovingly stages a family event with elaborate food and musicians, with the camera and lighting creating an intimate atmosphere and positive feelings. Such scenes stand in stark contrast to moments in the film that are set outside, on London's seedy side. One locality appears twice (20 min into the film and at 1:30): a narrow alleyway leading down to the river that the Russian mob habitually uses to dump corpses in the Thames. In particular during the second scene, the steep brick walls, anonymous and industrial environment, and slanting and slippery stone steps, dangerously evoking the risks of any missteps, are reminiscent of German expressionist cinema and as such contribute to the film's overall ghostly atmosphere. It is here that Nikolai, accompanied by Anna, convinces Kirill not to drown the baby, whose DNA will eventually betray Semyon's rape to the police. The echoing acoustics in this shot, along with Kirill's obvious emotional stress, visually underlined by his acting and acoustically through the orchestral non-diegetic music, dramatize this moment, which furthermore includes a somewhat non-heteronormative family unit that toys with Kirill's obvious attraction to Nikolai. The quiet intimacy of the scene contrasts sharply with the strong winds that tussle the actors' hair, suggesting audible diegetic ambient sounds that nevertheless do not interfere with the whispered dialogue in which Nikolai invites Kirill "to go home", addressing the very tension between homeliness and scary alienation that is at the center of Freud's uncanny. The melodic flute solo in the score that ends the scene as Anna holds the baby in her white cover, forming a stark contrast both to her immediate environment and the voices of the two mobsters on their way to party, underlines the tension in this scene and resonates with the film's overall spectral quality.

The spectrality of Cronenberg's film has already been discussed in a fascinating essay by Elizabeth Bronfen, whose focus, however, lies more on the contemporary political and economic debate. She writes that "In *Eastern Promises* we thus also have a specter on the loose, a phantomatic symptom of capitalism, giving voice to the destruction inherent to all creative economic takeovers" (Bronfen 2008, p. 63). For Bronfen, Cronenberg's film speaks to what she terms "The Violence of Money", a kind of economic world order that refutes any nostalgic desires for earlier forms of capitalism. I would like to take this

discussion one step further and instead discuss the ghostly presences in Cronenberg's film as a belated return of the repressed, as a moment of *Nachträglichkeit* in the political unconscious that reconnects the post-Wall debate to the contemporary discussion of the cosmopolitan and the convivial, to debates shaped by various recent publications (see Gilroy 2004; Spencer 2011).

This notion of the belated return also puts in an explicit visual appearance in the film during the scene when the body that was dumped in the scene mentioned above washes up on the shore near the Thames Barrier, the massive technological protection system that is to prevent the city from being flooded during extreme weather systems. As the film cuts to this scene, one of the rare moments that actually feature daylight, the camera follows the detective sent to solve the murder in what appears to be a hand-held tracking shot that quickly gives way to steady close-ups that focus on the disfigured corpse. As the senior officer explains about the prison tattoos to his junior colleague, viewers learn about some crucial aspects of the Russian mafia and their insignia, information that not only comes in handy as the film's plot further develops but also demonstrates that, underneath the clothes visible to passers-by on the street, London citizens may well reveal narratives not easily decoded by their neighbors. This secret form of information adds another uncanny layer to how London appears as a city whose population, coming from such different parts of the world, does not always share the same cultural heritage. Looked at in retrospect, Cronenberg's film already addresses many of the issues that many years later would be employed in the narratives that drove many voters towards voting for Brexit, turning the film in itself into an uncanny and somewhat clairvoyant gesture.

### 3. The Mob as Colonial Phantom

It is worth pointing out, maybe, that Derrida's *Spectres of Marx* is not built on a nostalgic desire—or the melancholia diagnosed in Gilroy's assessment of the postcolonial moment for European colonizing countries—to re-instate either the Cold War or the Eastern European political reality then current with all its secretive power structures, panoptic sense of surveillance, and general sense of indemnity on the side of the state apparatus. Rather, in a somewhat uncanny echo of Jürgen Habermas's writing on the unfinished project of modernity, Derrida's evocation of Marx's ghost serves the surprisingly conservative and simultaneously politically progressive purpose of reminding his contemporaries of the as-yet unfulfilled project of oppositional political thinking and of adversarial self-positioning vis-à-vis the unchallenged forces of free-market ideologies in the immediate post-Wall period. Derrida traces the specters of Marx to all those instances of power that object social structures to the selfish interest of those already privileged. The obligation to listen to these specters, as Cronenberg does in his film, follows from the realities of contemporary life and culture. This is how Derrida establishes that moral obligation:

> "At a time when a new world disorder is attempting to install its neo-capitalism and neo-liberalism, no disavowal has managed to rid itself of all of Marx's ghosts. Hegemony still organizes the repression and thus the confirmation of a haunting. Haunting belongs to the structure of every hegemony." (Derrida 2006, p. 46)

Derrida here draws on Antonio Gramsci's writing about hegemony, which defines the devious manner in which some cultural discourses manage to superimpose their logic onto those who in fact are the victims of that very discourse. This process of willful brain-washing constitutes a self-imposed return to a moment in Europe's intellectual history that predates the Kantian Enlightenment and as such justifies Derrida's claim that hegemonic systems are always the homely grounds of specters. On a more political note, the passage just quoted equates the social reality of post-Wall Europe to a hegemonic system of ideological control, to a reality, that is, which continues to call for a response inspired by the philosophical tradition started by Marx.

For Cronenberg's film, the situation is slightly different. *Eastern Promises* is hardly a call to political arms, nor does it constitute a philosophical rebirth of the suppressed. It instead works as a reminder that those dynamics supposedly buried in the rubble of the

Berlin Wall have not only travelled beyond the Iron Curtain but have effectively managed to claim a second life in the West. Cronenberg betrays no interest in the global questions over the economics of political ideologies. Yet, in the figure of the Russian mob, the very debate instigated in Derrida's book returns on the suppressed level of organized crime, of illegal systems of control, and of the bio-politics of human trafficking, forced prostitution, and contract murder. The Eastern Promises of the film's title sound uncomfortably hollow: they allude to the great failure of the post-Wall years of living up to the promises made in that first glow that followed the supposed conclusion of world history's endgame. As the West somewhat complacently celebrated its triumph over the Evil Empire, the rationale driving that empire merely went underground, spawning near-invisible offspring in the metropolitan centers of the capitalist ideology. Cronenberg's mobsters personify this move through their uncanny existence as the familiar figure of the local restaurant owner, charmingly portrayed by a music-loving Müller-Stahl, who functions as the ruthless godfather of a criminal organization that subscribes to the political logic of socialist regimes while simultaneously employing a capitalist logic of exploitation and financial gain that turns a blind eye to the suffering of those who produce the extra value extracted from their labor. By staging the localities connected to the mob in lush colors and through cozy associations, the film creates clear contrasts between appearances and ethical values. Its aesthetics clearly signal to the film's audience that material appearances are not always to be trusted. The Russian mob in *Eastern Promises* thus appears as the uncanny blend of the two forces that battled during the Cold War in that it embodies and cultivates the worst tendencies found in both systems but doing so only behind facades of respectabilities and through ostentatiously subscribing to conservative family values.

The Russian mob in the film accordingly exists only on a ghostly level: its activities play out in the backyards of London streets, at night-time, or behind closed doors. The seeming indemnity with which the clan leaders decide about the lives of others runs parallel to those features of political systems run awry that, in the West, people continue to associate with the pre-Wall East. The conflict between political and economic models supposedly laid to rest in 1989 thus returns as the specter of Russian crime. On more than one level, then, one could see Cronenberg's *Eastern Promises* as a modern adaptation of Bram Stoker's *Dracula* (1897). The novel famously reinserted the not-quite-human figure of the undead Count into London, thus confronting the enlightened West with its suppressed Other, both in terms of its ethnic self-image and the hypocrisy of its Puritanical sexuality, and the novel has also been read as an uncanny subtext that clandestinely addresses Ireland's clandestine history of colonial repression (see Smart and Hutcheson 2007). Cronenberg's vampires come in the shape of clean-cut and well-dressed businessmen, whose well-covered tooth-marks of prison tattoos are only visible and comprehensible to the initiated. In both works, the external threat figures as a return of the suppressed, whose supposed *Nachträglichkeit* only reminds its witnesses of the simultaneity and presence of its actual appearance. The film is, then, a fitting re-appropriation of the spectrality evoked in Derrida's comments on Marx. Cronenberg's cinematic play with history proves highly relevant, even as the historical contexts change: his 2007 film not only uncannily predates the global political events that would haunt the world just one year later, when the collapse of the American real estate market sent the global economy into a tailspin; it also resonates quite forcefully with the Brexit movement and its aftereffects of empty supermarket shelves and long lines outside gas stations. That political and economic processes once promoted to provide wealth and comfort at times return with a vengeance and wreak havoc is one of the messages that *Eastern Promises* presents through its spectral gesture towards phantasmagoric returns.

Derrida's book, in fact, was written before the developments started by the collapse of the Lehmann Brothers banking company, whose double identity in and of itself evokes spectral visions of doppelgängers. The 2008 collapse of the global banking order now increasingly resembles a moment that strangely embodied the ghostly return of a non-capitalist phantom. Derrida's spectral intervention thus appears uncannily clairvoyant. It reminds readers of how political and social ideologies are always already undead, how

they may at times exist below the threshold of human perception but eventually return to haunt the descendants of those who caused their original demise. In other words, from a ghostly perspective, it makes perfect sense for Cronenberg to set his film precisely in London, one of the global centers of a neoliberal capitalist economy, as that economy, to follow Fukuyama, killed off its rival sibling. How a post-Brexit London will develop its own sense of ghostly relations with both its colonial and now its former European past remains to be seen.

The hegemonic energy of that neoliberal world order did experience its own return of the repressed in that global collapse of the financial world order. When Derrida argued in the early 1990s that "haunting belongs to the structure of every hegemony", he envisioned a moment when the spectrality of the past would break through the surface of political and social realities. This breaking-through is what Cronenberg already envisioned in *Eastern Promises*. The film stages the conflicts between liberal and totalitarian political systems as playing out not in the grand arenas of the international summit circuit or, in other words, around the famous public and governmental sights of a city like London, such as the Houses of Parliament or Buckingham Palace, but instead at the rather seedy street corners that, for most people, create the very neighborhoods where they live. The film thus echoes other cinematic representations of London such as Mike Leigh's darkly dystopian *Naked* (1993), which also relies visually on "deserted streets, obscure alleyways, and an empty office block" (Mason 2001, p. 244). While Communism may have failed in Eastern Europe, its practice is alive and well, on a ghostly lever, in our midst, a place about which Joseph Conrad once had his narrator Marlow note in *A Heart of Darkness* (1899), "And this also [ . . . ] has been one of the dark places of the earth" (Conrad 2002, p. 105).

When *Heart of Darkness* projected the horror of colonialism back unto the London nightscape, it forced its British readers to reflect upon the legitimacy of its imperial ambitions. *Eastern Promises* invites comparisons that run along similar tracks, contrasting capitalist London with its underbelly of illegal and exploitative capitalism. The specters of both Marx and the mob raised by Cronenberg equally demand a moral response. The project Derrida sets out therefore describes in its exordium a task that, in order for humanity to live and prosper, every human being has to engage in. Derrida describes what is required for this exercise in his usual, circumspect ways, arguing that we need

> "to learn to live *with* ghosts, in the upkeep, the conversation, the company, or the companionship, in the commerce without commerce of ghosts. To live otherwise, and better. No, not better, but more justly. But *with them*. No *being-with* the other, no *socius* without this *with* that makes *being-with* in general more enigmatic than ever for us. And this being-with specters would also be, not only but also, a *politics* of memory, of inheritance, and of generations." (Derrida 2006, p. xviii)

It is, as Derrida rushes to add, by speaking about justice that he turns to specters, to the ghostly memories of past generations, and to the obligation that inheritance caries for those following in the future. Elsewhere, in a conversation with Michel Wieviorka, Derrida (Derrida and Wieviorka 2000) toys with the phonological similarities between amnesty and amnesia, again belaboring the ethical point that true forgiving cannot occur through an utter forgetting. The attachment to a specter thus figures as an injunction against its full reinstatement. In raising the specter of the mob in *Eastern Promises*, Cronenberg issues a dark reminder that even in one of the most liberal cities of the world, the open society is undermined by ghostly organizations built around violence and crime.

## 4. Traumatic Spectrality

Lest I be accused of doing the impossible, of fixing a specter, of pinning down an ephemeral ghost, of giving a final meaning to the inexorably elusive phantom of cinematic signification, allow me to offer a second angle at which one could discuss Cronenberg's *Eastern Promises* as a film haunted by beings whose metaphysical presence refuses any final explanation. This second ghost in the filmic machine (reminiscent of Arthur Koestler's dystopian comments on technological advances) differs from its spectral doppelgänger in

that, unlike its avatar, it does not relate to cultural forms of memory but instead haunts an individual as a result of past traumatic experiences (Koestler 1967). Both ghosts, however, form an uncanny alliance in that they ultimately relate back to the same historical developments. This second, traumatic, ghost appears in Anna's memories and experiences of her own family. As the daughter of a British–Russian couple, she grew up with an absent father, the child of a single mother whose solitary marital status, the film suggests, is the result of a failed cultural model of integration that caused her cross-cultural affair to falter, smoothly aligning with the film's overall tendency to oscillate "between phallic paternal *jouissance* and maternal desire" (Loren 2014, p. 155).

Anna's experience of her own life, then, is marked by a feeling of guilt: she sees her existence as the outcome of a wrongful romantic and sexual contact, the illegal offspring of an illicit romance. Those who have seen the film may wonder from what information I draw these conclusions: after all, we do not learn about these childhood memories directly. However, as with all true moments of trauma, it is through repetition that the psychic force of a wound prevents the sufferer from moving beyond the unconscious memories of the past, not allowing them to abreact the blocked affect. Therefore, when Anna's own affair with a colleague from work fails, she (like her Ukrainian uncle) probably blames that failure, and the subsequent loss of her unborn child, to the incriminating act of miscegenation, to her falling in love with a man whose African background makes of him a revenant of her absent father and his ethno-cultural strangeness. (It is only telling, given the mechanism of repression, that her attachment to her own father lives on in her mode of transportation, a Russian motorcycle.) When her uncle gives voice to the racist sentiments that haunt her own family romance, she gets very upset and leaves the room, her bodily reaction acting out the desire to flee from that explanation as it would force her to face her earlier memories.

In the psychoanalytic writings of Nicolas Abraham and Maria Torok, this kind of delayed, cross-generational surfacing of past traumatic injuries is described in more details. They suggest that "The presence of the phantom indicates the effects, on the descendants, of something that had inflicted narcissistic injury or even catastrophe on the parents" (Abraham and Torok 1994, p. 174). Derrida's specter as an effect of collective traumas thus returns as Abraham and Torok's phantom, as the kind of cross-generational haunting that concretizes, in all its phantasmagoric insubstantiality, a kind of personal memory that stands as a deferral of traumatic experiences. These repressed memories then surface, in the analytical frame developed by Abraham and Torok, as phobias who appear in people as "either a fear whose actual victims are their parents or, alternatively, a fear that the parents themselves had inherited and now transmit willy-nilly to their own reluctant offspring" (Abraham and Torok 1994, p. 181). Anna's anxieties about ethnic forms of belonging, about racist implications alive in her contemporary London environment, as well as the further trauma of her own unsuccessful pregnancy, all contribute to her own commitment to finding out about the story that attaches to the orphaned baby whose diary she is using as a pathway back into her family story. The film smoothly connects her personal biographical trauma with the violent biography of the orphaned baby's mother and the omnipresent, albeit clandestine, violence perpetrated by the mob members. All these diverse narratives are connected to matters of migrancy and legality in that many of those affected by these traumatic narratives suffer because of their status as foreign work force with questionable legal status. The way their status relates to their experiences of trauma the film connects to "the emergence of global oligarchies" that benefit from "the transnational flow of labor" and lead to "a breakdown of assumptions about national identity, citizenship, and the legal status—and therefore human rights—of workers" (Mihailovic 2013, p. 152). As forms of labor become somewhat ghostly in a reality that has moved from Cold War Manicheanism to global capitalism, Cronenberg's film addresses the fall-out that results from promises that all too frequently remain unfulfilled. Trauma becomes the consequence of such promises as handed out to Europeans who are trying to escape the logic of Eastern European political systems, or so *Eastern Promises* suggests. Addressing similar dynamics, film scholar Jeffrey Skoller has shown how cinema, in particular avant-garde film, has

found means of representing "traumatized histories of catastrophic events" (Skoller 2005, p. xliii) through the kind of spectral aesthetics that also feature in Cronenberg's film about the Russian mob in London.

## 5. Conclusions

Such a traumato-topical analysis of *Eastern Promises* views the film as a failed family romance, a film that allows for the romantic only through tropes of loss and failure. It is only appropriate, then, that the romance that slowly builds between Anna and Nicolai over both the technological ghost of her absent father and the rescue mission that saves the orphaned baby who is herself the product of a murderous mobster rape never quite takes off. Drawing on such "crucial moments of recognition and reversal" (Freeland 2012, p. 25), the film clearly invites its viewers to sympathize with Nicolai, whose secretive existence as an undercover agent only transpires very late in the film. However, his cool masculinity and gangster charm, adeptly represented by Viggo Mortensen's method acting, as well as his framing as an object of erotic desire—most explicitly in a violent nude scene at a public bath that required extensive rehearsals (Freeland 2012, p. 32)—turn him into the love object of not just Anna but also Kirill, whose homosexuality the film only cautiously attaches to the spectral existence of the Russian mob, thereby following a general tendency that Scott Loren has described when he notes "that gender is mutable" (Loren 2014, p. 152) in Cronenberg films. Just like viewers never quite know what to make of Nicolai's deep convictions, his sexual identity is variously and uncannily refracted, as when he kisses Anna in a frame that ironically stages the core family through an iconic representation of loving parents attached to their newborn child, despite the fact that any family connections between this triad are highly unlikely (but see Dunlap and Delpech-Ramey, who see in the film a turn towards "the ordinary life of couples"; Dunlap and Delpech-Ramey 2010, p. 333). Viewers of course know about the sad biographical beginnings of this baby, who has furthermore escaped mere seconds earlier a cruel drowning at the hands of Kirill, who, in turn, has quickly moved from being emotionally overwhelmed by the expectation to kill a baby to being enthusiastic about celebrating his supposed new role in the mob with Nicolai. Viewers also know that the romance between Anna and Nicolai has few chances of developing into anything substantial. This ghost of an affair—something that Cronenberg apparently feels may require a sequel to follow through (Freeland 2012, p. 26) —will surely disintegrate instantaneously, leaving Anna with yet another memory of a Russian phantom, another lost lover and potential father figure, thus further adding to her childhood trauma. The final scene, with Anna having adopted the baby girl and feeding her a bottle of milk while her mother and uncle provide a surrogate family environment in the background, finally allows for some sunlight to break through the noir aesthetics of the film's otherwise rainy and dark setting. However, the scene is set in a backyard that is anything but an open landscape, and *Eastern Promises* thus closes with an image of love restored, of parental tenderness and care, yet one whose narrow confines and precarious existence is more than palpably felt by the audience. Anna's brush with violence, her brave encounter with the mob, had led her—and those around her, like her uncle, who at one point is spirited away by Nicolai—to the brink of death and destruction, and the ghosts of the underworld are anything but abated.

Yet specters should be treasured, as Derrida suggests; or at least their effects should be acknowledged, if only in order to keep them at bay. Truly enough, at least Anna's personal ghosts seem to have been laid to rest, as the film's closing shots reveal. With the Russian mob now under close surveillance from the inside, with Nicolai implanted as a spectral informer, and with Anna's personal trauma at least momentarily inactive, a more idyllic future is evoked, albeit one that is exclusively female across three generations (Loren 2014, p. 168); until, as we suspect, some specter will return from its haunting grounds. Cronenberg's film indeed ends with a notion of ghosts as essentially undead, as entities always able and willing to return. What *Eastern Promises* raises in this way is not merely the specter of Russian crime as it infiltrates the liberal and democratic western

sphere: it invites its viewers instead to reflect on the fact that, all too frequently, people, businesses, and narratives are a mere front for a dark underworld whose ghostly existence lurks right under the surface. In *Eastern Promises*, Cronenberg employs the figure of the phantom as a means to address the very dynamic of Freudian repression and of the general notion of the unconscious with all its aspects of desire and morality. However, unlike in films like *A Dangerous Method*, Cronenberg here places London and its ethnic scenes of cross-border cultural identities at the center. The film makes visible that the promises of post Cold War political enthusiasm were not fully kept, that interior tensions remain, and that the materialism of capitalist politics all too frequently brings with it elements of violence and exploitation. In alluding to such topics, albeit via spectral aesthetics, Cronenberg's film also addresses a range of political issues that remain more than relevant for post-Brexit England as well, a place that continues to be haunted by its unkept promises.

**Funding:** This research received no external funding.

**Conflicts of Interest:** The author declares no conflict of interest.

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
