# Peer review of "Specters of Mob in David Cronenberg’s Eastern Promises"

_humanities, doi:10.3390/h10040116_

Round 1

Reviewer 1 Report

This is an engaging manuscript that applies spectrality, specifically the work of Derrida (among others), to David Cronenberg's film Eastern Promises in a manner that is sophisticated and evocative. The film itself is textually rich and provides much opportunity for the author to mine the hauntedness of the past (the phantom of colonialism, trauma, Eastern European political reality, etc.) in the present in contemporary London. 

There are aspects that would strengthen the piece, related to  argument, essay structure and prose.

In terms of argument and structure, the manuscript would benefit from a more balanced approach to incorporating research material. Some sections were very dense in terms of theorisation and segues to literary fiction (not all may be warranted, so as to avoid widening the parameters of the essay too much), which became apparent when discussions of the film seemed to have very little scholarly support and may come across as narrative description. By balancing discussion of the film and theory/concepts, this will produce a more seamless argument.

While the author has discussed various plot events in Eastern Promises, the argument would be more compelling by working more closely with specific scenes (and the aesthetics of the scene). This deep dive would be compelling reading, especially for film and media studies scholars. Additionally, there appears to be some issues that were mentioned briefly but need further elaboration e.g. migration. Tighter parameters around the various points raised in the manuscript may require some of these ideas be jettisoned in order to produce a deep, critical engagement with the main thesis.

Given the discussion of the uncanny and London, I would have liked to see a greater exploration of the spectral nature of London itself and how the aesthetics of the film (neo-noir) facilitate a reading of hauntedness.

The Introduction needs significant rewriting to explicate the argument that the author makes in this essay; at present the Introduction seems to discuss the other ways one might approach Cronenberg's film but which are not the foci of this essay. Section 2 'Ghosts of the Past, Ghosts of the Present' is where the author gets stuck into what their manuscript is really about.

The manuscript ends rather abruptly, without a clear conclusion.

Dotted throughout the manuscript are convoluted sentences that stretch over many lines, which are difficult to read. This can be easily remedied by having tighter sentences and/or breaking up particularly long sentences.

This manuscript has a lot of potential. If the necessary revisions are made, it would make a significant contribution to the scholarship.

Author Response

Dear editors, dear outside readers,

Many thanks for your careful reading and highly useful feedback on my essay!

For this reworked version of my essay, I have included the following steps:

  • include further research on the film, with a focus on gender aspects
  • offer more direct discussion of film scenes
  • reframe the introduction so that it more clearly situates my own essay in existing Cronenberg discourse
  • rework some of the longer sentences into smaller units
  • add a clearer conclusion and outlook
  • address some of the film's parallels to post-Brexit London and UK 

Reviewer 2 Report

This is a very neatly and tightly argued review of the film that makes some salient points in terms of neo-liberal Europe and some contemporary discourses around Britain's place post-Brexit (I did think perhaps some of the contemporary resonances might have been drawn out a little more, as uncanny reflections of the present in a near 15 year old film). It clearly explores these points through their representation in Eastern Promises and its Derridean and psychoanalytical approach is relevant to the analysis of the film.

I have few areas where the article could be stronger (aside from a very quick final proof-read to catch the very small number of typos: on page 3, line 115 'on of' should be 'one of'; and page 5, line 225-6 Ireland should read Ireland's). The background literature on Cronenberg (on which there is plenty) could be stronger - I'd particularly want to see the 'plethora of interpretive approaches' more strongly referenced to situate the argument more broadly. One of those approaches is gender, which has been widely explored as key trope in Cronenberg's film, and in Eastern Promises (by Anna Pasolini and others), while, this might not be the main focus of the psychoanalytical approach taken here, I believe that the gendered dimension of the aspects of femininity and motherhood, which are strongly implicated in the article, should be accounted for in the analysis (even just to broaden the supporting literature).

Author Response

(The authors gave the same response as above.)
